# No evidence that human GIGYF2 interacts with GRB10: implications for human disease

Jung-Hyun Choi[1,2,3,]*, Israel Shpilman[1,2,]*, Niaz Mahmood[1,2,]*, Shaghayegh Farhangmehr[4,5], Ulrich Braunschweig[4], Nathalia Gomide Cruz[6], Jun Luo[1,2], Reese Jalal Ladak[1,2], Angelos Pistofidis[2], Jiannis Ragoussis[6], T Martin Schmeing[2], Seyed Mehdi Jafarnejad[7], Benjamin J Blencowe[4,5], Nahum Sonenberg[1,2]

GIGYF2 (growth factor receptor–bound protein 10 [GRB10]-interacting GYF [glycine–tyrosine–phenylalanine] protein 2) reduces mRNA stability and translation via microRNAs, ribosome quality control, and several RNA-binding proteins. GIGYF2 was first identified in mouse cell lines as an interacting partner with GRB10, which binds to the insulin receptor and the insulin-like growth factor receptor 1. Mutations in the human *GIGYF2* gene were reported in autism. In mouse models, *Gigyf2* mutations engender several diseases. It was therefore thought that the GIGYF2-associated disease in humans is caused by defective GRB10 signaling. We show here that GIGYF2 does not interact with GRB10 in human cell lines, as determined by co-immunoprecipitation and proximity ligation assays. The lack of interaction is explained by the absence of the critical GYF domain–binding PPGΦ sequence in the human GRB10 protein. These results contrast with the current understanding that a GIGYF2/GRB10 complex is associated with human disease via insulin receptor and insulin-like growth factor receptor 1 signaling and underscore alternative mechanisms responsible for the observed phenotypes associated with mutations in the human *GIGYF2* gene.

## Introduction

Mouse growth factor receptor–bound protein 10 (GRB10)–interacting GYF (glycine–tyrosine–phenylalanine) protein 2 (GIGYF2, also called TNRC15; trinucleotide repeat–containing gene 15 protein) was first identified as a GRB10-interacting protein through a yeast two-hybrid assay (Giovannone et al, 2003). The gene encoding the orthologous human GIGYF2 protein is located on chromosome 2q37. The human *GIGYF2* gene has a high pLI (probability of loss-of-function intolerance) score of 1 (https://gnomad.broadinstitute.org/), indicating robust genetic stability (Fuller et al, 2019). Mutations in the human *GIGYF2* gene are genetically linked to autism spectrum disorder (ASD) as the Simons Foundation Autism Research Initiative (SFARI) gene database (https://gene.sfari.org/) classifies mutations of *GIGYF2* as a category 1 (high confidence) risk factor for ASD (some of the mutations in *GIGYF2* gene are shown in Fig 1) (Iossifov et al, 2014; Lim et al, 2017; Rubinstein et al, 2018; Wang et al, 2020; Zhou et al, 2022).

GIGYF2 harbors several protein-binding motifs: 4EHP (eIF4E homologous protein or eIF4E2), DDX6 (DEAD-box helicase 6), and GYF (Fig 1). GIGYF2 interacts with 4EHP via a conserved N-terminal motif (YXYXXXXLΦ, where Φ is a hydrophobic amino acid) to repress the translation of selective mRNAs in different biological contexts, including mouse embryonic development, activation of an innate immune response upon viral infections, and binding of microRNAs (miRNAs) or RNA-binding proteins to mRNAs (Morita et al, 2012; Fu et al, 2016; Tollenaere et al, 2019; Xu et al, 2022; Ladak et al, 2024). The GYF motif interacts with partner proteins via the PPGΦ motif, where Φ is a hydrophobic amino acid (F/I/L/M/V) (Kofler et al, 2005). The PPGΦ-containing GIGYF2-interacting proteins function in a variety of cellular processes via the tri-nucleotide repeat–containing gene 6 proteins (TNRC6A, TNRC6B, and TNRC6C), which are subunits of the microRNA-induced silencing complex, or the ribosome quality control factor zinc finger protein 598 (ZNF598), or the RNA-binding protein tristetraprolin (Kofler & Freund, 2006; Morita et al, 2012; Fu et al, 2016; Tollenaere et al, 2019; Sobti et al, 2023).

GRB10, a member of the GRB7 family of signaling adapters (Daly, 1998), binds to and inhibits the catalytic activity of the insulin receptor (IR) and insulin-like growth factor 1 receptor (IGF-1R) (Liu & Roth, 1995; Pandey et al, 1995). GRB10-mediated inhibition of IR and IGF-1R activity is linked to cognitive impairments associated with diabetes in rodents (Ma et al, 2013). GRB10 in most mouse species

[1]Rosalind and Morris Goodman Cancer Institute, McGill University, Montreal, Canada    [2]Department of Biochemistry, McGill University, Montreal, Canada    [3]Department of Biochemistry, Chungbuk National University, Cheongju, Republic of Korea    [4]Donnelly Centre for Cellular and Biomolecular Research, University of Toronto, Toronto, Canada    [5]Department of Molecular Genetics, University of Toronto, Toronto, Canada    [6]McGill Genome Centre, Victor Philip Dahdaleh Institute of Genomic Medicine, McGill University, Montreal, Canada    [7]Patrick G. Johnston Centre for Cancer Research, Queen's University Belfast, Belfast, UK

Correspondence: niaz.mahmood@mcgill.ca; nahum.sonenberg@mcgill.ca
*Jung-Hyun Choi, Israel Shpilman, and Niaz Mahmood contributed equally to this work

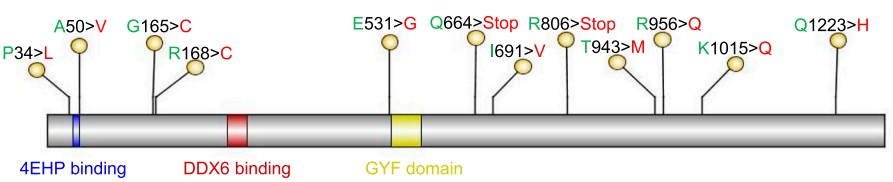

**ASD-related mutations in human GIGYF2**

**Figure 1. Key GIGYF2 mutations linked to autism.**
The different motifs of GIGYF2 are indicated: 4EHP stands for eIF4E homologous protein; DDX6 refers to DEAD (Asp-Glu-Ala-Asp)-box helicase 6; GYF stands for glycine–tyrosine–phenylalanine.

(Muridae) contains a PPGΦ motif, which mediates GRB10 interaction with GIGYF2 (Kofler et al, 2005). It was reported that mouse GIGYF2 potentiates the GRB10-mediated inhibition of IR and IGF-1R function (Giovannone et al, 2003; Xie et al, 2014). The SFARI database classifies GRB10 as a category 3 (suggestive evidence) risk factor for ASD, unlike GIGYF2, which is classified as category 1. Strikingly, the PPGΦ motif is absent in human GRB10 (Kofler et al, 2005). In this report, we show that, contrary to its designation as a GRB10-interacting protein, the human GIGYF2 does not bind to GRB10.

## Results

### The PPGΦ motif in GIGYF2 exists only in several species of the *Muridae* family

We examined GRB10 protein sequence alignments using Clustal Omega (Madeira et al, 2024) across different vertebrates for the presence of the PPGΦ motif (Fig 1). In addition, we performed a phylogenetic analysis using Randomized Axelerated Maximum Likelihood (Stamatakis, 2006), analyzing the nucleotide sequences corresponding to transcripts encoding GRB10 in various species (Fig S1). We noted that the PPGΦ motif and its conserved flanking region are present only in several species of the *Muridae* family (see Fig 2 for sequence alignment and Fig S1 for phylogenetic tree). To investigate the cause for the absence of the PPGΦ motif in human, we inspected the splicing pattern of orthologous *GRB10* genes in *Mus musculus* (house mouse; hereafter referred to as mouse) versus *Homo sapiens* (human). The PPGΦ motif is encoded in the alternatively spliced exon 5 of mouse *Grb10* (Fig 3A). In sharp contrast, the corresponding sequence in orthologous human *GRB10* is not predicted to be part of a coding exon. This is because both the donor and acceptor splice sites in the corresponding human sequence have diverged from the consensus splicing signals (5′ splice site GU changed to GC, and 3′ spice site AG diverged to GG, Fig 3B). As a result of mutations in the splice sites flanking the potential human exon, the open reading frame is shifted to introduce a premature stop codon (PTC) (in red, Fig 3B). To validate that the latter orthologous sequence is not included in the mature mRNA, we examined the splicing patterns of *GRB10* transcripts in mouse versus human. Reverse transcription–polymerase chain reaction (RT–PCR) was performed on RNA extracted from mouse brain cortex and human HAP1 (a near-haploid derived from the KBM-7 chronic myeloid leukemia cell line) and RPE1 (retinal pigment epithelial-1) cells, using a pair of primers designed to amplify the corresponding human and mouse loci (Fig 3C). Next, cycloheximide (CHX) treatment

was used to inhibit the nonsense-mediated decay (NMD) pathway to enable the detection of potentially spliced but unstable or transiently expressed transcripts. We found that the exon encoding the PPGΦ motif is predominantly included in the mouse brain cortex (442 base pairs), but not in the human cell lines (Fig 3D). As positive controls for CHX treatment and detection of NMD–affected isoforms, RT–PCR analysis of PTC-containing isoforms transcribed from the serine/arginine-rich splicing factor 2 (*SRSF2*) and *SRSF6* genes was performed (Fig S2). The data demonstrate that the PPGΦ motif is present in the predicted protein encoded by mouse GRB10 but absent from the orthologous protein in humans. This raised the critical question whether GIGYF2 binds to GRB10 in humans.

### Absence of detectable interactions between human GRB10 and GIGYF2 proteins

We first assessed the interaction between human GRB10 and GIGYF2 by performing co-immunoprecipitation (co-IP) experiments in human embryonic kidney 293T (HEK293T) cell extracts. All co-IP assays were conducted in the presence of ribonuclease A to eliminate potential RNA-bridging artifacts. Although, as expected (Morita et al, 2012), 4EHP and ZNF598 proteins co-precipitated with human GIGYF2, we failed to detect any co-precipitation of GRB10 and GIGYF2 (Fig 4A). Conversely, IGF-1R co-precipitated with human GRB10 as expected (Liu & Roth, 1995; Vecchione et al, 2003), but not with GIGYF2 (Fig 4B).

To exclude the possibility that the lack of co-IP of human GIGYF2 and GRB10 proteins was due to technical limitations or unforeseen artifacts of the co-IP assay, we used a proximity ligation assay (PLA). Corroborating the co-IP results, co-transfection of FLAG-GRB10 with v5-IGF-1R yielded robust PLA signals (Fig 4C and D). In contrast, very feeble signals were detected in cells co-transfected with FLAG-GRB10 and either v5-Empty or v5-GIGYF2 (Fig 4C and D). Taken together, these results indicate a lack of interaction between GRB10 and GIGYF2 in human cells.

### Human GIGYF2 and GRB10 proteins are components of distinct complexes

We next examined the distribution of GIGYF2 and GRB10 proteins in native complexes via size-exclusion chromatography (Superose 6) on lysates from human HEK293T cells. Human GIGYF2 and GRB10 proteins eluted in distinct fractions (Fig 4E). GIGYF2 primarily eluted in fractions associated with very high molecular mass complexes (between ~2 MD and > 660 kD) co-eluting with its known interactor TNRC6A (also known as GW182), a component of the microRNA-induced silencing complex (Sobti et al, 2023). In sharp contrast, GRB10 and IGF-1R predominantly

## Phylogenetic alignment of GRB10

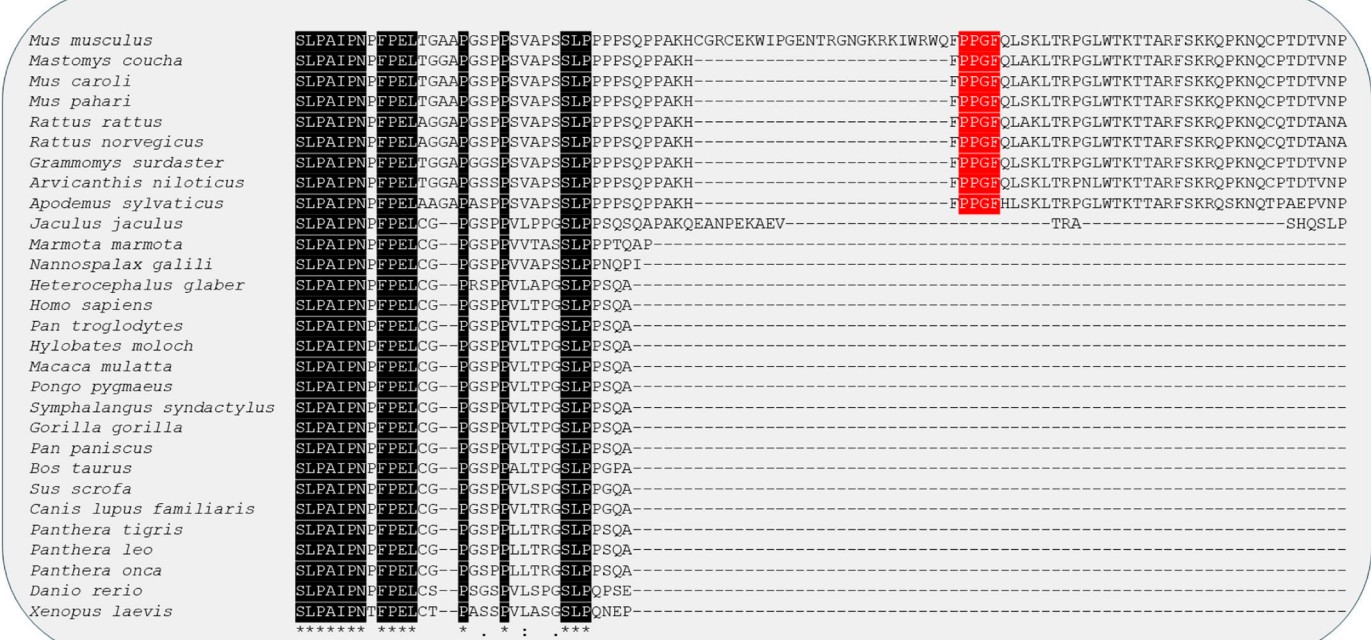

**Figure 2. PPGΦ motif of the GRB10 protein is present only in species of the Muridae family of rodents.**
Multiple sequence alignment of GRB10 proteins of different species. The alignment was performed using Clustal Omega (Sievers et al, 2011). Amino acids that are similar across all analyzed species are shaded in black. The PPG motifs are highlighted in red. The species names of the respective amino acid sequences are listed on the left. Amino acid sequences of GRB10 from a total of 29 species were analyzed for multiple sequence alignment. The binomial nomenclatures and common names (shown within brackets) are as follows: *Mus musculus* (House mouse), *Mastomys coucha* (Southern multimammate mouse), *Mus caroli* (Ryukyu mouse), *Mus pahari* (Gairdner's shrewmouse), *Rattus rattus* (Black rat), *Rattus norvegicus* (Brown rat), *Grammomys surdaster* (African woodland thicket rats), *Arvicanthis niloticus* (African grass rat), *Apodemus sylvaticus* (Wood mouse), *Jaculus jaculus* (Lesser Egyptian jerboa), *Marmota marmota* (Alpine marmot), *Nannospalax galili* (Middle East blind mole-rat), *Heterocephalus glaber* (Naked mole-rat), *Homo sapiens* (Human), *Pan troglodytes* (Chimpanzee), *Hylobates moloch* (Silvery gibbon), *Macaca mulatta* (Indochinese rhesus macaque), *Pongo pygmaeus* (Bornean orangutan), *Symphalangus syndactylus* (Siamang), *Gorilla gorilla* (Western gorilla), *Pan paniscus* (Bonobo), *Bos taurus* (Cattle), *Sus scrofa* (Wild boar/swine), *Canis lupus familiaris* (Dog), *Panthera tigris* (Tiger), *Panthera leo* (Lion), *Panthera onca* (Jaguar), *Danio rerio* (Zebrafish), and *Xenopus laevis* (African clawed frog).

co-eluted in fractions corresponding to lower molecular mass (<660 kD). The disparate elution patterns demonstrate that GIGYF2 and GRB10 are components of distinct protein complexes, consistent with divergent molecular functions.

AlphaFold3 generates high-quality models of protein complexes (Abramson et al, 2024). We submitted the protein sequences of human GRB10 and GIGYF2 to the AlphaFold3 server. As expected for noninteracting proteins, the five co-complex models (Abramson et al, 2024) had no commonality in the suggested GRB10: GIGYF2 interfaces, and each had very poor chain-pair interface–predicted template modeling scores (0.17–0.21) and minimum chain-pair–predicted aligned error scores (23.4–25.5). This indicates that AlphaFold3 does not predict the interaction between GRB10 and GIGYF2.

## Discussion

Our findings demonstrate that human GRB10 does not bind GIGYF2. This is an unsurprising finding considering the absence of the PPGΦ motif in human GRB10, which mediates the binding of the mouse GRB10 to GIGYF2 via its GYF motif (Giovannone et al, 2003). Specifically, an exon encoding the PPGΦ motif of GRB10 is predominantly included in *Muridae* but not in human mRNAs because of evolutionary changes in donor and acceptor splice sites that preclude its splicing in. Importantly, the exon sequence itself has diverged such that it harbors a PTC. Previous studies have estimated that ~50% of alternative exon differences between human and mouse are represented by such "genome-specific" exons, that is, exons that are spliced only in one of the two species because of gain or loss of the exon and/or critical cis-elements required for exon inclusion (Pan et al, 2005). Few such splicing changes have been functionally characterized and linked to important phenotypic differences (Xia et al, 2024), and as such, the possible biological implications of these differences are generally unclear, or else have been missed or overlooked in annotation efforts. In particular, the Dependency Map (DepMap) portal of the Broad Institute (Tsherniak et al, 2017) states that GIGYF2 might be involved in tyrosine kinase signaling. The STRING database (Szklarczyk et al, 2023), a comprehensive resource providing information on known and predicted interactions, describes GIGYF2 as a GRB10-binding protein (Fig S3). Considering our results, the entries in which GIGYF2 and GRB10 proteins are mentioned as interacting partners in human must be amended.

GRB10 inhibits IGF-1R and IR function independent of GIGYF2 (Morrione, 2000; Dufresne & Smith, 2005; Ma et al, 2013). GRB10 can form

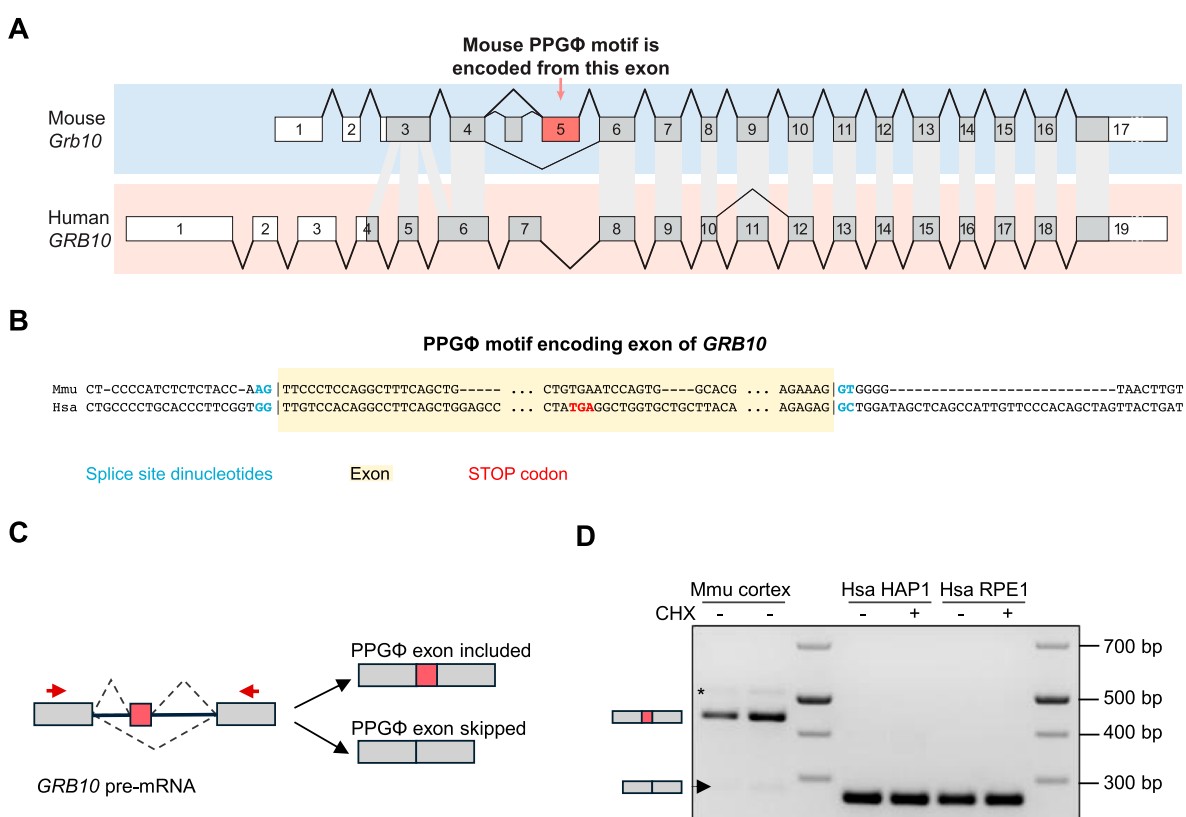

**Figure 3.  PPGΦ motif of the human GRB10 is not present because of an alternative splicing event.**
**(A)** Schematic representation of the structure of murine and human genes encoding GRB10. Gray shading indicates coding sequence (CDS); thick lines and exon numbering indicate the splicing pattern of the major isoform according to Matched Annotation from NCBI and EBI (MANE), whereas thin lines indicate alternative splicing. White exonic parts indicate untranslated regions (UTRs). Light gray–shaded areas indicate orthologous regions of exons. **(B)** Comparison of the PPGΦ motif encoding sequences in human (Hsa) and mouse (Mmu) genes encoding GRB10. The splice sites are indicated in blue, and the alternative exon in yellow. **(C)** Schematic model illustrates the design of a set of primers amplifying sequences of human and mouse *GRB10* genes flanking the exon encoding the PPGΦ motif. The exons and introns are represented by rectangles and lines, respectively. The exon encoding the PPGΦ motif is shown as a "red" rectangle, and flanking exons are colored in gray. Arrowheads depict the location of the forward and reverse primers used to detect the exon encoding PPGΦ motif in panel "(D)". **(D)** RT–PCR analysis of total RNAs derived from mouse cortex and CHX-treated human HAP1 and RPE1 cells using primers described in "(C)". The PCR products were resolved on a 2.5% agarose gel. The size of the skipped exon is 265 base pairs in humans and 277 base pairs in mice. The arrowhead indicates the skipped exon (minor band) in mice. The asterisk represents a band of unknown identity that is not the expected size for the target splice variant, which is only amplified in the samples of mouse origin.

a complex with an E3 ubiquitin ligase known as neural precursor cell–expressed developmentally down-regulated protein 4 (NEDD4), which promotes IGF-1R degradation (Monami et al, 2008). Studies on mouse models explored cognitive impairment as a complication of diabetes through the repression of IGF-1R by GIGYF2 (Xie et al, 2014). GIGYF2 was shown to cause insulin resistance in obese mice by impairing the phosphoinositide 3-kinase/protein kinase B (PI3K/AKT) pathway (Lv et al, 2024), which facilitates the translocation of glucose transporter GLUT4 from the cytoplasm to the cell membrane. Knock-down of mouse *Grb10* augments downstream signaling pathways through the IGF-1R–mediated phosphorylation of IR substrates, extracellular signal–regulated kinase 1/2 (ERK1/2), and AKT (Dufresne & Smith, 2005). Because human GRB10 cannot bind to GIGYF2, it is most probable that GIGYF2 and GRB10 function in humans through disparate pathways to inhibit the activity of IR and IGF-1R. Given these insights, further research on the relationship between GIGYF2 and GRB10 is necessary to clarify their role in insulin signaling and related pathologies. We recently described a mouse model in which 4EHP, the binding partner of GIGYF2, was depleted (Wiebe et al, 2020). The mice exhibited a

canonical ASD-like phenotype, including exaggerated hippocampal metabotropic glutamate receptor–dependent long-term depression (mGluR-LTD) and social behavior deficits (Wiebe et al, 2020). In another study, we demonstrated that the interaction between GIGYF2 and 4EHP is necessary for their mutual co-stabilization (Morita et al, 2012), as the loss of either protein results in the degradation of the other without affecting mRNA levels. Thus, the cause of ASD in humans could be explained by the dysregulation of miRNA-induced translational silencing mediated by 4EHP, because of changes in GIGYF2 rather than the GIGYF2/GRB10 axis. Our findings are consistent with the involvement of 4EHP/GIGYF2 in the etiology of autism in humans.

# Materials and Methods

### Phylogenetic analysis of the PPGΦ motif

To construct the phylogenetic tree, nucleotide sequences corresponding to transcripts encoding GRB10 from various species were

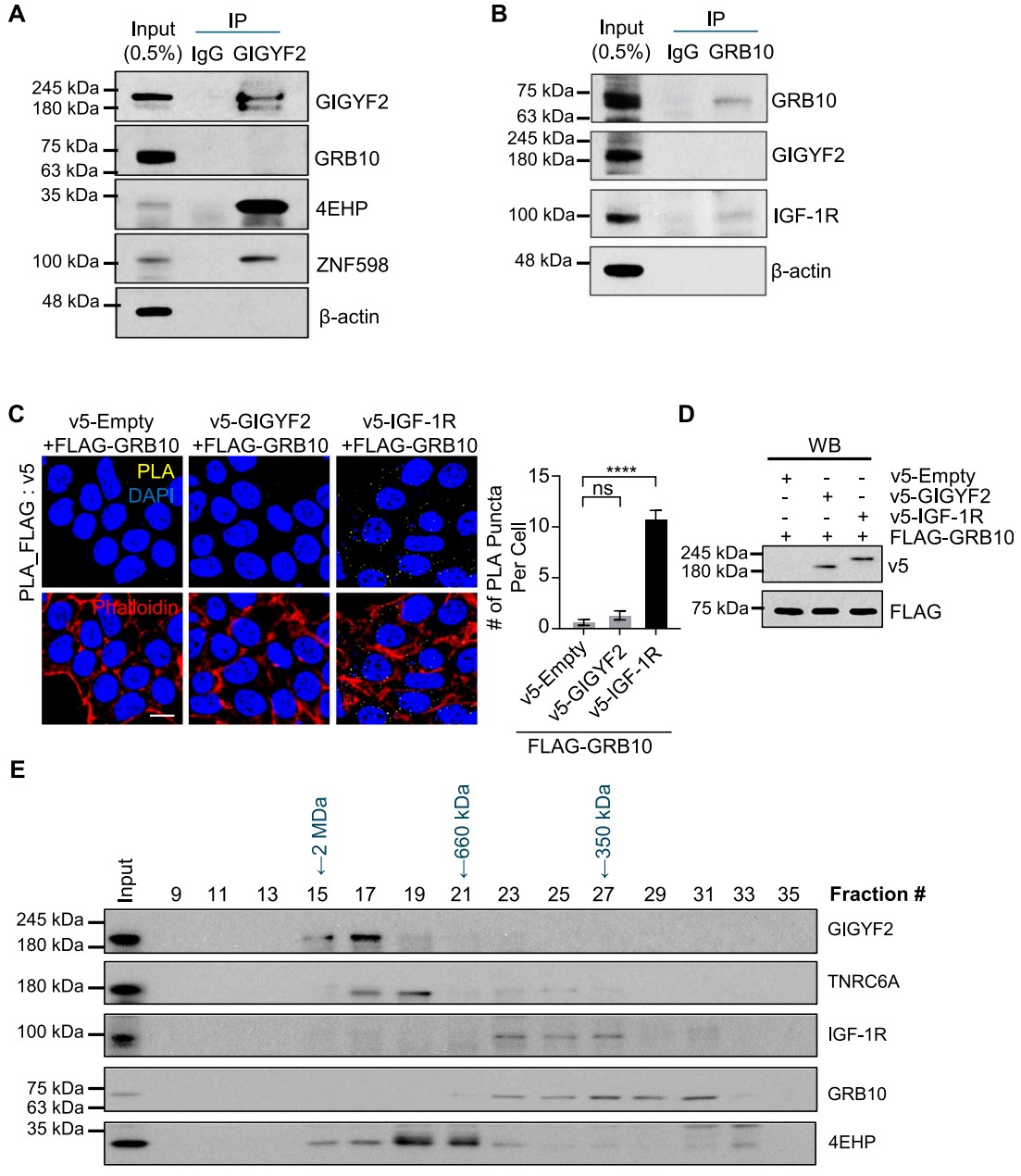

**Figure 4. Analysis of interactions between human GIGYF2 and GRB10 proteins.**
**(A)** co-IP analysis of endogenous GIGYF2 protein interactions in HEK293T cells. Immunoprecipitation was performed using an anti-GIGYF2 antibody, followed by Western blot analysis with the indicated antibodies. **(B)** co-IP analysis of endogenous GRB10 protein interactions in HEK293T cells. Immunoprecipitation was performed using an anti-GRB10 antibody, followed by Western blot analysis with the indicated antibodies. **(C)** PLA for detecting interactions between FLAG-GRB10 and v5-IGF-1R or v5-GIGYF2. PLA signals are shown in yellow; the nucleus and actin cytoskeleton were counterstained with DAPI and phalloidin, respectively. Scale bar = 10 μm. The bar graph represents the average number of PLA signals from at least 30 cells per sample (n = 3 independent experiments). **(D)** Western blot analysis of cell lysates from PLA shown in panel "(C)". **(E)** Analysis of endogenous GIGYF2- and GRB10-containing complexes by size-exclusion chromatography. A total of 10 mg of protein from HEK293T cells was loaded onto a Superose 6 column and run at a flow rate of 0.4 ml/min. Fractions of 0.5 ml were collected, and 50 μl of each fraction was analyzed by Western blotting.

aligned using Clustal Omega (Madeira et al, 2024). The resulting multiple sequence alignment was used to infer a maximum likelihood phylogenetic tree with Randomized Axelerated Maximum Likelihood (Stamatakis, 2006), executed via the T-REX web server (Boc et al, 2012), applying the GTRCAT substitution model with 25 discrete site rate categories and 100 rapid bootstrap replicates. Branch lengths and tree topology were optimized under the maximum

likelihood criterion. The final tree was visualized and annotated using Interactive Tree of Life (iTOL) v6 (Letunic & Bork, 2024).

## Cell line and cell culture

HEK293T (Thermo Fisher Scientific) cells were cultured in DMEM (319-005-CL; Wisent Inc.) supplemented with 10% FBS (S12450; R&D

Systems) and 1% penicillin/streptomycin (450-200-EL; Wisent Technologies). RPE1 cells were cultured in DMEM (D5796; Sigma-Aldrich) supplemented with 10% FBS (12483-020; Gibco), 1 mM sodium pyruvate (11360-070; Gibco), and 1% penicillin/streptomycin (15140122; Gibco). HAP1 cells were cultured in Iscove's Modified Dulbecco's Medium (12440053; Gibco) supplemented with 10% FBS (12483-020; Gibco) and 1% penicillin/streptomycin (15140122; Gibco). To inhibit translation, cells were incubated in media supplemented with either 50 $\mu$M CHX solubilized in DMSO (01810; Sigma-Aldrich) or DMSO alone for 6 h before lysis. Cells were incubated in a humidified atmosphere of 5% $CO_2$ at 37°C.

### Antibodies and plasmids

The following antibodies were used in the indicated dilution index: rabbit anti-GIGYF2 (1:3,000, A303-732A-M; Bethyl Laboratories), rabbit anti-GW182 (1:1,000, A302-329A; Bethyl Laboratories), rabbit anti-GRB10 (1:1,000, 3702; Cell Signaling), rabbit anti-IGF-1R (1:1,000, ab182408; Abcam), rabbit anti-eIF4E2 (1:3,000, GTX82524; Genetex), rabbit anti-ZNF598 (1:1,000, GTX119245; Genetex), mouse anti-$\beta$-actin (1:5,000, A5441; Sigma-Aldrich), mouse anti-v5 (1:3,000, ab27671; Abcam), mouse anti-FLAG (1:5,000, ab49763; Abcam), rabbit anti-FLAG (1:2,500, F7425; Sigma-Aldrich). The $\lambda$N-v5–tagged GIGYF2 vector was described before (Choi et al, 2024). v5-tagged-IGF-1R (#98344; Addgene) and FLAG-GRB10 (#37481; Addgene) were used for the PLA assay.

### RT–PCR

Total RNA was extracted from RPE1 and HAP1 cells using RNeasy Mini Kit (74106; QIAGEN) following the manufacturer's instructions and including a DNase digestion step (79254; QIAGEN). Briefly, cells were washed twice with PBS and directly lysed in RLT buffer supplemented with 2-mercaptoethanol (M3148; Sigma-Aldrich). Embryonic mouse cortices were harvested at E18.5, lysed in RLT buffer supplemented with 2-mercaptoethanol, homogenized with QIAshredder (79656; QIAGEN), and subjected to RNA extraction as above. RT–PCR was performed according to the manufacturer's instructions using a QIAGEN one-step RT–PCR kit (210212; QIAGEN) using 10 ng of total RNA for human cells and 20 ng of total RNA for mouse cortices. The primer sequences used for RT–PCR are listed in Table S1.

### Western blotting

HEK293T cells were lysed with radioimmunoprecipitation assay (RIPA) lysis buffer (89901; Thermo Fisher Scientific), supplemented with a cOmplete EDTA-free protease inhibitor (04693124001; Roche) and phosphatase inhibitor (P5726, P0044; Sigma-Aldrich) cocktails. Protein quantification was performed using the Bio-Rad Protein Assay Dye Reagent Concentrate (Cat# 500-0006). Equal volumes of extracts were mixed with 2X Laemmli sample buffer (Cat# 1610747; Bio-Rad), which included 5% $\beta$-mercaptoethanol. Proteins were denatured by heating at 95°C for 3 min and loaded onto a sodium dodecyl sulfate–10% polyacrylamide gel (SDS–PAGE) for electrophoresis. Proteins were transferred onto a nitrocellulose membrane, which was subsequently blocked using 5% milk in Tris-buffered saline with 0.1% Tween-20 (TBST, pH 7.6). Primary antibodies were applied to the membrane for 16 h. The membranes were washed three times with TBST and incubated with horseradish peroxidase–conjugated secondary antibodies (Bio-Rad) for 1 h. Signals were visualized using enhanced chemiluminescence (ECL-Plus; PerkinElmer, Inc.) against an X-ray film (Denville Scientific, Inc.).

### Size-exclusion chromatography

HEK293T cells were harvested by centrifugation and resuspended in 500 $\mu$l of lysis buffer containing 50 mM Tris–HCl (pH 7.4), 150 mM sodium chloride (NaCl), 1 mM EDTA (pH 8.0), and 1% Triton X-100. The lysis buffer was supplemented with a cOmplete EDTA-free protease inhibitor (04693124001; Roche) and phosphatase inhibitor cocktails (P5726, P0044; Sigma-Aldrich). A Superose 6 Increase HR 10/300 column (Cytiva) was pre-equilibrated with SEC (size-exclusion chromatography) buffer (25 mM N-2-hydroxyethylpiperazine-N-2-ethane sulfonic acid supplemented with potassium hydroxide [HEPES-KOH, pH 7.5], 150 mM NaCl, 0.5 mM tris[2-carboxyethyl] phosphine [TCEP]) at a flow rate of 0.4 ml/min. A total of 10 mg of protein from the HEK293T cell lysate was loaded onto the column. Fractions of 0.5 ml were collected, with 50 $\mu$l from each fraction used for Western blot analysis.

### Immunoprecipitation

Cells were washed with cold PBS and collected by scraping into lysis solution (40 mM HEPES-KOH [pH 7.5], 120 mM NaCl, 1 mM EDTA, 50 mM sodium fluoride [NaF], 0.3% 3-[(3-cholamidopropyl) dimethylammonio]-1-propanesulfonate [CHAPS]), supplemented with cOmplete EDTA-free protease inhibitor (04693124001; Roche) and phosphatase inhibitor (P5726, P0044; Sigma-Aldrich) cocktails. Pre-cleared lysates containing 2 mg of protein were incubated overnight at 4°C with 5 $\mu$g of anti-GIGYF2 or anti-GRB10 antibody, along with 40 $\mu$l of pre-washed Protein G agarose beads slurry (16-266; Millipore) and RNase A (EN0531; Thermo Fisher Scientific). After the incubation, the beads were washed three times for 10 min each using a wash buffer (50 mM HEPES-KOH, pH 7.5, 150 mM NaCl, 1 mM EDTA, 50 mM NaF, 0.3% CHAPS), supplemented with cOmplete EDTA-free protease inhibitor and phosphatase inhibitor cocktails. Proteins were eluted from the beads using the Laemmli sample buffer.

### PLA

PLA was performed using Duolink reagents (Sigma-Aldrich, DUO92101) following the manufacturer's guidelines. Cells were fixed in 4% PFA in sucrose for 15 min, followed by permeabilization with PBS containing 0.1% Triton X-100 for another 15 min. After permeabilization, the cells were blocked with Duolink blocking solution for 1 h at 37°C and then incubated overnight at 4°C with primary antibodies. After this, cells were washed with wash buffer A and treated with PLA probes for 1 h at 37°C. A ligation reaction was then carried out for 30 min at 37°C. The PLA signals were subsequently amplified using an amplification buffer for 100 min at 37°C. Finally, after a wash with wash buffer B, the samples were mounted

onto glass slides and examined using an Airyscan microscope (Zeiss).

## Co-complex model generation

Prediction of co-complexes was performed with AlphaFold3 via the AlphaFold Server (Beta) (https://alphafoldserver.com), using default seed autogeneration, which provides a set of five models (Abramson et al, 2024). Co-complexes modeled were of NP_001337743.1 (GRB10) and NP_001096617.1 (GIGYF2), and resulting model (.cif) and statstisiv (.json) files were analyzed with ChimeraX (Meng et al, 2023) and PyMOL (The PyMOL Molecular Graphics System, Version 3.0 Schrödinger, LLC).

## Statistical analysis

Statistical analysis was conducted using Prism 10 (GraphPad Prism Software Inc.). Error bars represent the SD from the mean of three independent replicates. Statistical significance was set at $P < 0.05$.

# Supplementary Information

# Acknowledgements

This work was supported by grants from the Canadian Institutes of Health Research (CIHR; 148423), Simons Foundation Autism Research Initiative (AR-DIR-00006609), and Norman Zavalkoff Family Foundation to N Sonenberg. Additional support was provided by CIHR project grants (480827 and 517998) to BJ Blencowe. TM Schmeing was supported by a CIHR Project Grant 178084. J Ragoussis was supported by Genome Canada Genomic Facility Funding Opportunity for Technology Development. J-H Choi was supported by a Conrad F. Harrington Fellowship. N Mahmood was supported by a post-doctoral fellowship from the Goodman–Kahvejian family. The authors thank John Burke for AlphaFold3 tips. N Sonenberg and TM Schmeing are members of the Centre de recherche en biologie structurale, funded by Fonds de Recherche du Québec (Health Sector) Research Centre Grant #288558.

## Author Contributions

J-H Choi: formal analysis, validation, investigation, visualization, methodology, and writing—original draft, review, and editing.

I Shpilman: data curation, formal analysis, validation, investigation, visualization, and writing—original draft, review, and editing.

N Mahmood: data curation, formal analysis, supervision, validation, investigation, visualization, and writing—original draft, review, and editing.

S Farhangmehr: formal analysis and investigation.

U Braunschweig: data curation, formal analysis, and investigation.

NG Cruz: data curation, formal analysis, and investigation.

J Luo: investigation.

RJ Ladak: investigation.

A Pistofidis: investigation.

J Ragoussis: resources, supervision, and writing—review and editing.

TM Schmeing: resources, data curation, formal analysis, supervision, and writing—review and editing.

SM Jafarnejad: resources and writing—review and editing.

BJ Blencowe: resources, supervision, and writing—review and editing.

N Sonenberg: conceptualization, resources, supervision, and writing—original draft, review, and editing.

## Conflict of Interest Statement

The authors declare that they have no conflict of interest.

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
