## [Reviewer comments · Life Science Alliance]

Life Science Alliance

No evidence that human GIGYF2 interacts with GRB10: implications for human disease

Jung-Hyun Choi, Israel Shpilman, Niaz Mahmood, Shaghayegh Farhangmehr, Ulrich Braunschweig, Nathalia Cruz, Jun Luo, Reese Ladak, Angelos Pistofidis, Jiannis Ragoussis, T. Schmeing, Seyed Mehdi Jafarnejad, Benjamin Blencowe, and Nahum Sonenberg

DOI: <https://doi.org/10.26508/lsa.202503334>

Corresponding author(s): Nahum Sonenberg, McGill University and Niaz Mahmood, McGill University

Review Timeline:

Submission Date:	2025-03-31
Editorial Decision:	2025-05-07
Revision Received:	2025-06-01
Accepted:	2025-06-05

Scientific Editor: Tim Fessenden

Transaction Report:

May 7, 2025

RE: Life Science Alliance Manuscript #LSA-2025-03334-T

Dr. Nahum Sonenberg
McGill University
Department of Biochemistry
1160 Pine Avenue West
Room 615
Montreal, PQ H3A 1A3
Canada

Dear Dr. Sonenberg,

Thank you for submitting your manuscript entitled "No evidence that human GIGYF2 interacts with GRB10: implication for human disease". This manuscript was evaluated by two expert reviewers, whose comments are below. As you will see, both reviewers find the work is timely, clear, and of significant interest to the field. We would be happy to publish your paper in Life Science Alliance pending the minor changes sought by Reviewer 1 and final revisions necessary to meet our formatting guidelines.

- Please upload your main manuscript text as an editable .doc file.
- Please add a Running Title and a Summary Blurb/Alternate Abstract in our system
- Please add ORCID ID for the corresponding secondary author- they should have received instructions on how to do so.
- Please add a Category and Keywords for your manuscript in our system.
- Please add the X and Bluesky handles of your host institute/organization as well as your own or/and one of the authors in our system.
- The titles in both the system and the manuscript file must be consistent with each other.
- Please add authors' contributions to our system as well.
- Please upload your Table in editable .doc or Excel format. It can be included at the bottom of the main manuscript file or sent as separate files.
- Please add molecular weight markers to the blots in Figure 4.

A. FINAL FILES:

B. MANUSCRIPT ORGANIZATION AND FORMATTING:

Sincerely,

Reviewer #1 (Comments to the Authors (Required)):

Mutations in the human GIGYF2 (Growth factor receptor bound protein 10 (GRB10)-interacting GYF (glycine-tyrosine-phenylalanine) gene were reported in autism spectrum disorder and Parkinson's disease. Studies in mouse and human cell lines reported an interaction between GRB10 and GIGYF2, which bind to the insulin receptor and insulin-like growth-factor receptor. Impairment of this interaction is thought to engender human disease. The authors report the striking finding that GIGYF2 does not interact with GRB10 in human cell lines, as determined by proximity ligation and co-immunoprecipitation assays. These results were further corroborated by size exclusion chromatography experiments. The lack of interaction is compellingly explained by the absence of the critical GYF-domain binding PPGΦ sequence in the human GRB10 protein. These results refute the accepted notion that a GIGYF2/GRB10 complex is associated with human disease via IR and IGF-1R tyrosine kinase signaling. The study should have a strong impact on research in the field, especially as it relates to possible drug development. The experiments were expertly performed, and the data are convincing. The paper is clearly written, albeit the English ought to be improved. The authors should consider the comments below to strengthen the paper.

Points:

1. The authors did not emphasize enough the importance of their findings vis-a-vis human disease and future drug development. In the absence of this publication the interaction between GRB10 and GIGYF2 could be potentially targeted to treat human disease based on the published studies. This could be a huge waste of time and resources,
2. The dramatic differences between the alternative splicing patterns of GRB10 pre-mRNA in mouse versus human is highly intriguing. The authors should discuss how rare is this occurrence in general. It is most important to know if such differences have some evolutionary or physiological implications. This ought to be addressed in the literature.
3. Fig. 3. The authors mention two human alternative splicing isoforms but show only one. Why? Also, in Fig.3A GRB 10 is spaced out, while in 3B and C it is not (GRB10).
4. Fig. 4. It is rather surprising that the IP of GRB10 and IGF1-1R is rather poor (<2%). This should be commented on.
5. Fig. 5. This is a very convincing result. While not essential, the GRB10-IGF-1R complex co-elutes as a 500 kDa (should be corrected to kDa, not KDa) complex. Have the authors probed for other predicted components of the complex?
6. The authors must improve the writing, which is inadequate.

"mutations in the gene encoding GIGYF2 exhibited disease phenotypes". How can mutations exhibit disease phenotypes? Mutations rather cause or engender disease phenotypes.

Minors:

- a. "Conversely, IGF-1R co-precipitated with human GRB10 as expected²¹, but no co-precipitation of the GIGYF2 protein (Fig. 4B)". Should be changed to "..but not with GIGYF2".
- b. "we used an orthogonal validation approach with a proximity ligation assay (PLA)". Should read "...we used a proximity..."
- c. "fractionate in common complexes"- what is the meaning of "common"?

Reviewer #2 (Comments to the Authors (Required)):

GIGYF2 was found as interacting protein with GRB10. GIGYF2 mutations are linked to autism spectrum disorder, and it was thought that this is due to disruption of GIGYF2-GRB10 interaction. However, as shown unequivocally in this manuscript the human GRB10 lacks the motif required for binding GIGYF2. Several reports showed that GIGYF2 also interacts with 4EHP. This interaction competes with the binding of eIF4E and inhibits mRNA translation. Thus, raising the possibility that GIGYF2 mutations interfere with the binding of 4EHP resulting in elevated mRNA translation, which is consistent with increased protein synthesis observed in ASD. Overall the results are convincing. It will strengthen the manuscript if the authors show 4EHP in the size-exclusion chromatography (Fig. 4E).

We thank the reviewers for their constructive comments, which have greatly improved the manuscript. We appreciate that they recognized the importance of our work. In the revised version of the manuscript, we have included additional information on the implications of the results for future studies related to GRB10 and GIGYF2 in the discussion. We hope that the reviewers find the new improved version acceptable for publication. We added 2 authors who performed a phylogeny search. Our responses to the reviewers' comments are as follows:

From the journal:

Along with points mentioned below, please tend to the following:
-Please upload your main manuscript text as an editable .doc file.

Response: As per the requirement of the journal, we have uploaded the editable .doc file.

-Please add a Running Title and a Summary Blurb/Alternate Abstract in our system

Response: We have now included a running title (on Page 1) and a summary blurb (on Page 2).

-Please add ORCID ID for the corresponding secondary author- they should have received instructions on how to do so.

Response: The ORCID ID is now added.

-Please add a Category and Keywords for your manuscript in our system.

Response: We have now added the category and provided the keywords for the manuscript.

-Please add the X and Bluesky handles of your host institute/organization as well as your own or/and one of the authors in our system.

Response: We have provided the X handles of the institute and the lab.

-The titles in both the system and the manuscript file must be consistent with each other.

Response: We ensure that the titles are consistent with each other.

-Please add authors' contributions to our system as well.

Response: We have added the authors' contributions to the system.

-Please upload your Table in editable .doc or Excel format. It can be included at the bottom of the main manuscript file or sent as separate files.

Response: The editable table is added in the main manuscript as .doc file (at the bottom of the manuscript on Page 13)

-Please add molecular weight markers to the blots in Figure 4.

Response: We have now included the molecular weight markers to the blots in Figure 4.

Response: As per the requirement of the journal, we have now included a summary blurb.

Reviewer #1 (Comments to the Authors (Required)):

1. The authors did not emphasize enough the importance of their findings vis-a-vis human disease and future drug development. In the absence of this publication the interaction between GRB10 and GIGYF2 could be potentially targeted to treat human disease based on the published studies. This could be a huge waste of time and resources,

Response: In response to the reviewer's request, we have expanded our discussion to emphasize the significance of the findings and their implications for the design of future research.

2. The dramatic differences between the alternative splicing patterns of GRB10 pre-mRNA in mouse versus human is highly intriguing. The authors should discuss how rare is this occurrence in general. It is most important to know if such differences have some evolutionary or physiological implications. This ought to be addressed in the literature.

Response: In response to the reviewer's request, we have provided a more detailed discussion on the rarity of such splicing events in the revised manuscript.

3. Fig. 3. The authors mention two human alternative splicing isoforms but show only one. Why? Also, in Fig.3A GRB 10 is spaced out, while in 3B and C it is not (GRB10).

Response: Human GRB10 has several predicted variants, none of which contain the PPGF motif encoding region. We showed NM_001350814 as an example of human GRB10 in Fig. 3A because it is the major human isoform, as determined by our interrogation of the 'Matched Annotation from NCBI and EMBL-EBI' (MANE). We reconfirmed that there is no space between GRB and 10 in 3A.

4. Fig. 4. It is rather surprising that the IP of GRB10 and IGF1-1R is rather poor (<2%). This should be commented on.

Response: GRB10 and IGF1-1R interact with many proteins. These interactions are dynamic and transient. It is thus not an unreasonable result.

5. Fig. 5. This is a very convincing result. While not essential, the GRB10-IGF-1R complex co-elutes as a 500 kDa (should be corrected to kDa, not KDa) complex. Have the authors probed for other predicted components of the complex?

Response: As mentioned by the reviewer, we also thought that having GRB10-IGF-1R as a positive control was sufficient for this experiment, and therefore not probed for other predicted components

of the complex. However, we have now included a blot with 4EHP which is in the same complex as GIGYF2.

6. The authors must improve the writing, which is inadequate. "mutations in the gene encoding GIGYF2 exhibited disease phenotypes". How can mutations exhibit disease phenotypes? Mutations rather cause or engender disease phenotypes. Response: In response to the reviewer's request, we have edited the sentence in the following manner: 'In mouse models, *Gigyf2* mutations engender several diseases.'

Minors:

a. "Conversely, IGF-1R co-precipitated with human GRB10 as expected²¹, but no co-precipitation of the GIGYF2 protein (Fig. 4B)". Should be changed to "..but not with GIGYF2".

Response: In response to the reviewer's request, we have edited the sentence in the following manner: '*Conversely, IGF-1R co-precipitated with human GRB10 as expected (Vecchione et al, 2003), but not with co-precipitation of the GIGYF2 protein*'

b. "we used an orthogonal validation approach with a proximity ligation assay (PLA)". Should read "...we used a proximity..."

Response: In response to the reviewer's request, we have edited the sentence: '*To exclude the possibility that the lack of co-precipitation of human GIGYF2 and GRB10 proteins was due to technical limitations or unforeseen artifacts of the co-IP assay, we used a proximity ligation assay (PLA)*'

c. "fractionate in common complexes"- what is the meaning of "common"?

Response: In response to the reviewer's request, we have now edited the sentence.

Reviewer #2 (Comments to the Authors (Required)):

GIGYF2 was found as interacting protein with GRB10. GIGYF2 mutations are linked to autism spectrum disorder, and it was thought that this is due to disruption of GIGYF2-GRB10 interaction. However, as shown unequivocally in this manuscript the human GRB10 lacks the motif required for binding GIGYF2. Several reports showed that GIGYF2 also interacts with 4EHP. This interaction competes with the binding of eIF4E and inhibits mRNA translation. Thus, raising the possibility that GIGYF2 mutations interfere with the binding of 4EHP resulting in elevated mRNA translation, which is consistent with increased protein synthesis observed in ASD. Overall the results are convincing. It will strengthen the manuscript if the authors show 4EHP in the size-exclusion chromatography (Fig. 4E).

Response: We appreciate the reviewer's positive feedback on the manuscript. According to the reviewer's suggestion, we have now included a blot for 4EHP in the size exclusion chromatography (Fig. 4E). We hope that the reviewers find the new improved version acceptable for publication.

June 5, 2025

RE: Life Science Alliance Manuscript #LSA-2025-03334-TR

Dr. Nahum Sonenberg
McGill University
Department of Biochemistry
1160 Pine Avenue West
Room 615
Montreal, Quebec H3A 1A3
Canada

Dear Dr. Sonenberg,

Thank you for submitting your Research Article entitled "No evidence that human GIGYF2 interacts with GRB10: implications for human disease". It is a pleasure to let you know that your manuscript is now accepted for publication in Life Science Alliance. Congratulations on this interesting work.

DISTRIBUTION OF MATERIALS:

Again, congratulations on a very nice paper. I hope you found the review process to be constructive and are pleased with how the manuscript was handled editorially. We look forward to future exciting submissions from your lab.

Sincerely,
